# In-sensor reservoir computing system for latent fingerprint recognition with deep ultraviolet photo-synapses and memristor array

Zhongfang Zhang[1], Xiaolong Zhao [1]✉, Xumeng Zhang [2]✉, Xiaohu Hou[1], Xiaolan Ma[1], Shuangzhu Tang[2], Ying Zhang[1], Guangwei Xu[1], Qi Liu[2] & Shibing Long [1]✉

Detection and recognition of latent fingerprints play crucial roles in identification and security. However, the separation of sensor, memory, and processor in conventional ex-situ fingerprint recognition system seriously deteriorates the latency of decision-making and inevitably increases the overall computing power. In this work, a photoelectronic reservoir computing (RC) system, consisting of DUV photo-synapses and nonvolatile memristor array, is developed to detect and recognize the latent fingerprint with in-sensor and parallel in-memory computing. Through the Ga-rich design, we achieve amorphous $GaO_x$ (a-$GaO_x$) photo-synapses with an enhanced persistent photoconductivity (PPC) effect. The PPC effect, which induces nonlinearly tunable conductivity, renders the a-$GaO_x$ photo-synapses an ideal deep ultraviolet (DUV) photoelectronic reservoir, thus mapping the complex input vector into a dimensionality-reduced output vector. Connecting the reservoirs and a memristor array, we further construct an in-sensor RC system for latent fingerprint identification. The system maintains over 90% recognition accuracy for latent fingerprint within 15% stochastic noise level via the proposed dual-feature strategy. This work provides a subversive prototype system of DUV in-sensor RC for highly efficient recognition of latent fingerprints.

Fingerprint recognition is the preferred technique in biometrics which measures and analyzes the characteristics of human identity[1–3]. In most cases, latent fingerprints are not visible to the naked eye and require visualization[4,5]. The extraction and recognition of fingerprints play critical roles in personal identification, criminal investigation, and biometric security[6–8]. The quasi-zero background on the Earth's surface and the strong absorption of deep ultraviolet (DUV) waveband by organic residues endow DUV light with a promising capability to detect and recognize latent fingerprints[9–11]. Nevertheless, the conventional ex-situ DUV fingerprint recognition systems feature separated sensors, memory, and processor, which seriously deteriorate the latency of decision-making and inevitably increase the overall computing power (Fig. 1a)[12–14]. In addition, these systems utilize additional optical filters for charge-coupled devices (CCDs) and complementary metal-oxide-semiconductor (CMOS) image sensors, increasing the complexity of the entire system for latent fingerprint identification[15–17]. Therefore, to

[1]School of Microelectronics, University of Science and Technology of China, Hefei, China. [2]Frontier Institute of Chip and System, Fudan University, Shanghai, China. ✉e-mail: xlzhao77@ustc.edu.cn; xumengzhang@fudan.edu.cn; shibinglong@ustc.edu.cn

**Fig. 1 | Reservoir computing system based on photo-synapse and memristor device array. a** Data processing mode of traditional fingerprint recognition system, realized by independent optical sensor, memory chip and processor. **b** Schematic of the human visual recognition system comprising the retina, optical neurons, and visual cortex in the human brain. **c** Proposed RC system with optical synapses as the input layer of the reservoir and the memristor device array as the readout network.

The inset in the dashed box is the abstract photoelectronic RC system. The original optical information is transmitted into the photoelectronic reservoir, where the inputs are nonlinearly mapped into feature outputs based on the PPC effect. And then, the memristor array receives the outputs of the reservoir and implements readout training.

simplify the system construction and enhance the processing efficiency of the DUV fingerprint recognition system, it is urgent to develop a new principle device and new computing architecture.

Inspired by the biological sensory systems, neuromorphic architectures bring innovative thoughts to improve the efficiency of DUV fingerprint recognition because of their in-sensor and in-memory computing merits, relying on artificial neural network (ANN) computational models. Among various ANN models, reservoir computing (RC) only needs to train the output layer of the network and is demonstrated to be suitable for processing complex spatiotemporal data with the lowest computational cost[18]. Recently, emerging devices have been developed for RC system to condense or process spatiotemporal information with higher energy efficiency beyond CMOS technology[19–21]. However, these RC techniques are based on electrical stimuli and thus require complex circuits to support the information transmission between the sensor and RC system, which induces extra energy consumption and latency. Fortunately, a promising strategy of in-sensor RC based on optoelectronic devices has been proposed for temporal sensory information processing and verified with the assistance of system simulation[22,23]. In order to fulfill the in-sensor applications, the optoelectronic devices should be marked by the properties of nonlinearity response, short-term memory (STM), multiple states and stability. Nevertheless, the waveband utilized in above works is not suitable for DUV detection. Ultra-wide bandgap semiconductors with their bandgap immediately corresponding to the DUV region, provide subversive scheme for filter-free DUV sensors[24–28]. Especially, gallium oxide is currently regarded as the most promising material when compressively considering the DUV sensitivity, physicochemical stability, workability, availability, and cost[29–31]. Moreover, its amorphous counterpart (a-GaO$_x$) exhibits obvious persistent photoconductivity (PPC) effect[32,33], catering to the implementation of DUV in-sensor RC system. Therefore, designing an a-GaO$_x$ device with enhanced PPC effect and performing system demonstration is crucial to promote the DUV in-sensor RC system for latent fingerprint recognition.

In this work, we constructed a DUV in-sensor RC system based on photo-synapses and memristor array, mimicking the biological visual systems, for the highly efficient recognition of latent fingerprints. By Ga-rich component design, we implemented a-GaO$_x$ DUV sensors with enhanced PPC effect that illustrates nonlinearly adjustable conductivity, enabling the synaptic features, such as paired-pulse facilitation (PPF) and STM. These features ensure the a-GaO$_x$ photo-synapse as an expected DUV photoelectronic reservoir, which maps the complex input vector into a dimensionality-reduced output vector. Besides, the analog memristor array stores the weights of the output layer and enables parallel in-memory computing of feature outputs from the photo-synapses. As a result, we demonstrate the inputting, mapping, featuring, training, and recognition of fingerprint images based on a hardware DUV in-sensor RC system. By adopting a dual-feature strategy, the recognition accuracy of fingerprint images maintains over 90% even under 15% noise level, demonstrating the robust anti-noise characteristics of the system. This DUV in-sensor RC system will make a difference for highly efficient latent fingerprint recognition in the future for criminal investigation and security.

## Results

### DUV in-sensor RC for latent fingerprint recognition

Inspired by the biological visual systems (Fig. 1b), a DUV in-sensor RC system was configured for high-efficiency latent fingerprint recognition based on photo-synapses and memristor array (Fig. 1c). In the human visual perception system, sense and transmission of external stimuli rely on parallel networks of receptors (retina), neurons, and visual cortex in the brain, making the compact system efficient for solving complex and unstructured real-world problems. The visual signals are received by the retina and then transmitted along neurons and synapses to the visual cortex that carries out memory, learning, and recognition functions for further processing. Therefore, different from

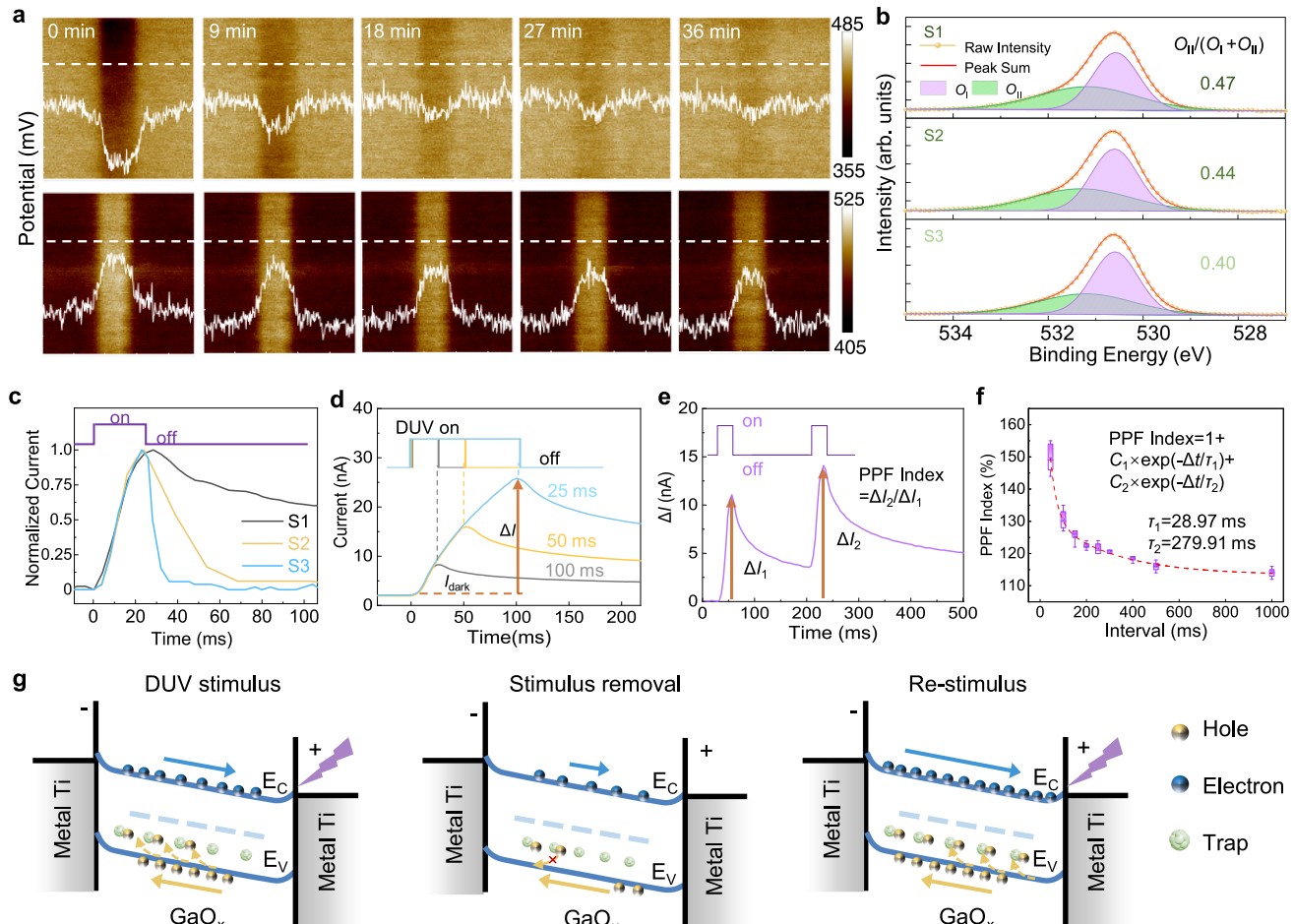

**Fig. 2 | PPC effect and synaptic behavior of the a-GaO$_x$ DUV sensor. a** KPFM results of the a-GaO$_x$ surface after injecting electrons (top panel) and holes (bottom panel) by negative/positive biasing. The inset in each image indicates the potential profile along the white dashed line. **b** XPS spectral fitting of the O1$s$ peak of the a-GaO$_x$ film with different oxygen contents (S1, S2, and S3). **c** The normalized photoresponse $I$-$t$ curves of the a-GaO$_x$ films with different oxygen contents (S1, S2, and S3). **d** The macroscopic PPC and volatile STM effects observed in Ga$_2$O$_3$ photo-toelectronic device under various DUV pulse widths (25 ms, 50 ms, and 100 ms).

The response current $\Delta I$ is the difference between the maximum photocurrent and the initial dark current. **e** PPF behavior of the a-GaO$_x$ photo-synapse induced by the PPC effect. Taking $\Delta I_1$ as the response current of the first stimuli and $\Delta I_2$ of the second, the PPF index is calculated as the ratio of $\Delta I_2$ to $\Delta I_1$. **f** Dependence of PPF indexes varying with pulse intervals from 45 ms to 1000 ms, fitted by a double-exponential function. Each pulse interval contains 10 measurements. **g** Influence of oxygen-vacancy-related traps on the generation, recombination, and regeneration processes of photo-generated carriers in the a-GaO$_x$ photo-synapse.

the traditional architecture for latent fingerprint recognition, the bio-inspired RC architecture is capable of sensing and processing in parallel so as to ensure high efficiency and low power dissipation[34].

The RC system basically consists of a reservoir layer and output readout layer, as shown in the dashed box of Fig. 1c. In an RC system, it is required that the states of the reservoir should be nonlinearly mapped by the temporal inputs and correlate to their former states[35], the weights of the readout layer are trained for specific learning tasks, and the final outputs are normally based on a linearly weighted combination of the reservoir feature outputs. The PPC effect with nonlinearly tunable characteristics renders the a-GaO$_x$ photo-synapses as ideal photoelectronic reservoirs, as shown in the left panel of Fig. 1c. Memristors feature excellent adjustable resistive state characteristics and naturally perform the vector-matrix multiplication in an array structure, making it suitable to serve as readout layer[36–38], as shown in the right panel of Fig. 1c. Therefore, we built an in-sensor RC system for latent fingerprint recognition based on DUV photo-synapse and memristor array. The fabrication processes of the photo-synapse and details of memristor array are shown in the Methods section and Supplementary Information.

## Carrier trapping in a-GaO$_x$ film and synaptic characteristics of the a-GaO$_x$ DUV sensor

In our previous work[39], the PPC effect in a-GaO$_x$ film has been suppressed by a high-temperature annealing process to implement high responsibility and swift response speed for high-speed DUV sensor application. But, for photo-synapse application, the PPC effect, which ensures the dynamically modulated conductivity, should be taken full advantages to mimic the synaptic plasticity[40,41].

To verify the mechanism of the PPC effect in the fabricated a-GaO$_x$ film, Kelvin probe force microscopy (KPFM), which uncovers the potential distribution of the object surface, has been performed with negative and positive biasing (Supplementary Fig. 1). In the target regions (5 × 5 μm²), only the middle parts (1 × 5 μm²) were injected with extrinsic electrons or holes. After the carrier injection, the dynamic evolution of both the surface potential distributions and the relative height in the target regions were measured every 9 min as presented in Fig. 2a. The recovery of target-region potential in both situations takes a long time, indicating the abundant electron and hole traps inside the a-GaO$_x$ film. Since the potential recovery of the hole-injected region lags obviously behind the situation of electron injection, the trapping effect of holes rather than electrons contributes more to the persistent conductivity.

The typical calculation results demonstrate that the oxygen vacancies, one of the intrinsic deficiencies in $Ga_2O_3$, could act as deep donors[42–44], and dominate the trapping phenomenon of none-quilibrium carriers. In view of the potential associations between the oxygen vacancy and hole trapping, we fabricated a-$GaO_x$ DUV sensors with different O contents in the film, labeled S1, S2, and S3 (see Methods section). X-ray photoelectric spectroscopy (XPS) characterizations of O1s peak of S1, S2, and S3 are performed as shown in Fig. 2b. The first peak $O_I$ is attributed to the lattice oxygen, and the second peak $O_{II}$ is associated with the oxygen vacancies[32,40,41]. The highest $O_{II}$/$O_I$ ratio demonstrates the most abundant oxygen vacancies in the as-sputtered a-$GaO_x$ S1 film with the lowest oxygen content. To testify the effect of this component design, normalized time-dependent photo-response curves were measured under a light pulse illumination of 25 ms, as shown in Fig. 2c. Obviously, the a-$GaO_x$ sample S1 with the highest $O_{II}$ component exhibits the most significant PPC effect. Therefore, the deliberately enlarged PPC effect by Ga-rich design turns the sample S1 into an ideal photo-synapse. All following experiments are based on the sample S1.

To further estimate the feasibility of nonlinear response characteristics as a photoelectronic reservoir, the PPC effect of this a-$GaO_x$ photo-synapse has been studied systematically. Figure 2d shows the time-dependent photoresponse curves of the a-$GaO_x$ photo-synapse under different light pulse widths (25 ms, 50 ms, and 100 ms). The decay process of the current after the removal of external light stimuli reveals the enhancement of the PPC effect with increasing the light pulse width. Concurrently, the intrinsic volatile characteristics of the photocurrent imply the STM effect of the device. Obviously, a longer illumination time leads to a higher reinforced PPC effect. Defining the response current ($\Delta I$) as the difference between maximum photo-current and initial dark current, it always exhibits nonlinear dynamics with the light pulse under various power densities, as shown in Supplementary Fig. 2. Besides, it should be noted that a negative voltage pulse strategy on the back gate was utilized to release the extra photocurrent caused by the PPC effect (Supplementary Fig. 3), to ensure the repeatable operation of the device[45].

In a biological neural system, the PPF as one typical function of the synapse, demonstrates the ability to process continuous temporal information. To validate the PPF effect, we applied consecutive light stimuli on the device, exhibited in Fig. 2e. During operation, the bias was set at 1 V, the power density of DUV pulses was fixed at 450 nW/cm², and the pulse width was fixed at 25 ms. Herein, a time-dependent photoresponse curve is depicted, revealing the real-time modulation of the channel conductivity or synaptic weight. Even though the power density was the same for both stimuli, the $\Delta I$ of the latter stimulus was obviously higher than that of the former, indicating the facilitation effect of synaptic plasticity. Taking $\Delta I_1$ as the response current of the first stimulus and $\Delta I_2$ of the second, the PPF index, as a ratio of $\Delta I_2$ to $\Delta I_1$, is calculated to be 128.4% in this case. To demonstrate the dependence of PPF on pulse interval, we then tested and extracted the PPF index under various intervals, as shown in Fig. 2f. The results show that the PPF index decreases exponentially with increasing pulse intervals, while it keeps almost independent of the power density of the light pulse (Supplementary Fig. 4). Moreover, the PPF index versus pulse interval curve can be fitted by a double-exponential function:

$$PPF\ index = 1 + C_1 \exp\left(-\frac{\triangle t}{\tau_1}\right) + C_2 \exp\left(-\frac{\triangle t}{\tau_2}\right) \quad (1)$$

where $C_1$ and $C_2$ are the facilitation ranges, and $\tau_1$ and $\tau_2$ with fitted values of 28.97 ms and 279.91 ms are the characteristic time constants relative to a rapid and slow relaxation time, respectively[46]. As a result, short pulse interval approaching to rapid relaxation constant exhibits higher PPF index or facilitation efficiency, making the STM generated by different inputs more distinguishable. In addition, the

photoresponse of the a-$GaO_x$ photo-synapse could always maintain nonlinearity even under multi-bit pulse stimuli (Supplementary Fig. 5), which plays an important role in classification and recognition tasks in biological activities[20,47]. Therefore, this nonlinear relationship between conductivity and external stimuli of the photo-synapse has the potential to construct a feature space for nonlinear mapping. The above-mentioned basic neuromorphic characterizations of the a-$GaO_x$ device reveal its promising potential for serving as a photoelectronic reservoir.

Figure 2g visualizes in detail the generating and trapping/detrapping behavior of nonequilibrium carriers inside the a-$GaO_x$ photo-synapse. Under DUV light stimulus (left panel), the photo-generated electron-hole pairs increase in the active layer, while the holes drift toward the electrode/$GaO_x$ interface and get captured by oxygen-vacancy-related traps. After stimulus removal (middle panel), the energy barrier in the detrapping process prolongs the annihilation of photogenerated current[48]. Thus, during re-stimulus process (right panel), the accumulated trapped holes lay a footstone of increasing free electron concentration and lead to the PPF effect.

**Nonlinear mapping of 4-bit inputs of the a-$GaO_x$ DUV reservoir**
The feature extraction of the original image simplifies the recognition process and improves the efficiency[49]. Owing to the nonlinear PPF of synaptic plasticity, the inputs sequence of the a-$GaO_x$ photo-synapse reservoir can be distinctly mapped into feature outputs. To assess the capability of the feature mapping of the reservoir, we perform the measurement of a 4-bit optical stream, which can be mimicked by the corresponding 4-bit inputs within "0000" to "1111", as shown in Fig. 3a. Each periodical input waveform (25 ms pulse width, 45 ms pulse interval) is considered as one bit, in which the "off" and "on" state of the light pulse denote "0" and "1" in the basic binary image.

The configuration of the feature space of inputs/outputs is the basis of readout training. Therefore, all the $I$-$t$ characteristics of all 4-bit inputs of pixel sequences have been measured and sampled for feature values. To illustrate the feature sampling, the $I$-$t$ curves of three representative inputs of "0001" (in blue), "0011" (in red), and "1101" (in purple) of the a-$GaO_x$ reservoir are exhibited in Fig. 3b. Although the last pulses are all "1", their decay processes after the input sequences are different. Therefore, the final state of the reservoir not only relates to the last input, but also depends on its real-time state, indicating the lateral connections in such an a-$GaO_x$ reservoir[22,23]. Based on the conspicuous difference, each pixel sequence can be featured by current sampling to realize feature extraction. For accurate feature extraction, sampling point 1 (SMP1) is defined as the average current value in the sampling time (ST) section after a sampling delay (SD) from the last pulse. Obviously, different SD and ST conditions will lead to discrepant SMP1 results, thus the optimal sampling section needs to be determined.

The SMP1 value has a negative correlation with SD and ST due to the fading effect of response current, as shown in Fig. 3c. Based on a statistical analysis of 100 cycles for all input sequences, the reservoir outputs with SD ranging from 0 to 20 ms and ST from 10 to 30 ms show similar fluctuations for SMP1 evolution, as shown in Supplementary Fig. 6. Apparently, the situation of 0 ms SD and 10 ms ST presents a relatively larger SMP1 value, which will facilitate the distinction of all 16 input sequences. Therefore, this sampling condition is utilized for current sampling and feature space configuration throughout the following experiment, as shown in Fig. 3d. In addition, similar statistical results (100 cycles for each input) based on 20 stochastically selected devices further validate the reliability and repeatability of the a-$GaO_x$ reservoir (Supplementary Fig. 7). Since there are potential overlaps between SMP1 distributions of two certain inputs (e.g., "1011" and "1101" in the dashed circle of Fig. 3d), a dual-feature strategy, namely adding an additional SMP2 at the end of the 3rd pulse region (see the dashed square), has been proposed to sharpen the

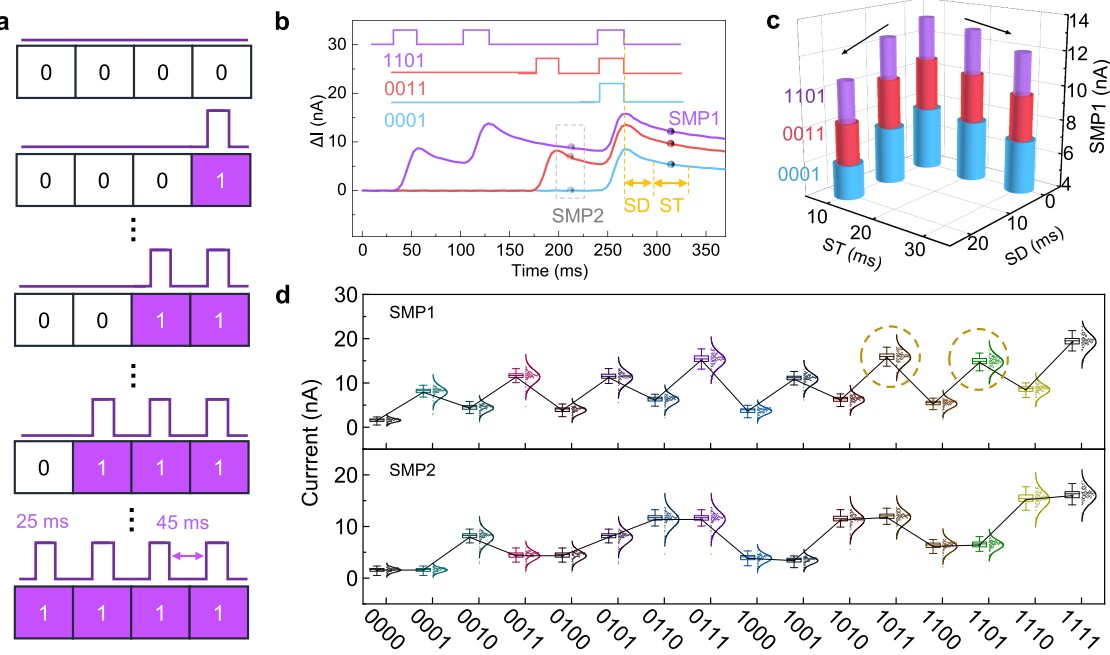

**Fig. 3 | Nonlinear mapping of 4-bit inputs based on the a-GaO$_x$ reservoir. a** 4-bit inputs in the form of a binarized number and equivalent light pulses from "0000" to "1111". The pulse width is 25 ms and pulse interval is 45 ms here. **b** *I-t* photo-response characteristics and input-output feature extraction strategy of three representative inputs of "0001", "0011", and "1101". **c** SMP1 versus increasing SD and ST of the three representative inputs. Owing to the fading memory, the SMP1 value has negative correlation with SD and ST. **d** The statistical results of the dual-feature strategy (SMP1 and SMP2) in 100 repetitive cycles for the 16 inputs, depicted by a box error diagram and normal distribution.

output features. Consequently, the feature space based on nonlinear photoresponse configures the classification process of the reservoir, reducing the dimensionality of raw data from 4-bit digital inputs to 2 analog outputs that serve as the inputs of the linear readout layer[50,51]. The energy consumption per pulse operation of the photoelectronic reservoir can be estimated to be $E = 20\,\text{nA} \times 1\,\text{V} \times 25\,\text{ms} = 0.5\,\text{nJ}$, indicating that the reservoir architecture possesses potential energy-efficient characteristic.

### Fingerprint recognition with fully-hardware DUV in-sensor RC system

To verify the feasibility of DUV in-sensor RC for fingerprint recognition, we constructed a hardware system, composed of a photosynapse reservoir layer and a memristor readout layer, as shown in Fig. 4a. The relationship between the mathematical model and the physical hardware of this system has been illustrated (see Supplementary Fig. 8). In such a system for DUV fingerprint recognition, the images are first converted into DUV light pulses. And then, the pulse signals are projected parallelly onto the a-GaO$_x$ photo-synapse reservoir layer, generating single or dual feature outputs. After receiving the outputs of reservoirs, a memristor array loading pre-trained weights performs the in parallel in-memory readout process. In the RC system, only the weights in the readout layer need to be trained, reducing the training costs for DUV fingerprint recognition. A photograph of the local hardware test board system with sensor array, memristor array, and data interfaces is shown in Supplementary Fig. 9. To perform the training and recognition of fingerprint images, 40 fingerprints of 5 people (8 for each) are chosen from the Second Fingerprint Verification Competition (FVC 2002) database (Supplementary Fig. 10). Preprocessing of the raw human fingerprint images includes cropping, scaling, binarizing, and rejoining to 20 × 4 pixels in order to adapt the 4-bit light pulse input as shown in Fig. 4b. Correspondingly, the reservoirs (20 a-GaO$_x$ photo-synapses)

generate 20 pairs of feature outputs (SMP1 and SMP2) per image for the training of the memristor array network. By simulating the readout network training as shown in Fig. 4c, the dual-feature strategy successfully achieves 100% accuracy after 100 training epochs, demonstrating a much faster convergence rate than in the single-feature situation (500 epochs). Thus, a dual-feature strategy is employed, and only a dimensionality-reduced 40 × 5 weight matrix needs to be trained for each fingerprint image. As an example, a recognition accuracy for unseen fingerprint images has been simulated to be around 92% based on this dual-feature strategy, where the expanded sample amounts of the fingerprint images ensure the high recognition accuracy (see Supplementary Fig. 11).

The memristor device with analog conductivity is deliberated to enable the hardware realization of a fully connected readout network (the characteristics of the memristors are shown in Supplementary Fig. 12). As exhibited in Fig. 4d, e, both the colormaps and statistical histogram of the 200 weights obtained by software simulation and memristor hardware verification after the training are highly consistent within tolerable error. Therefore, this hardware DUV in-sensor RC system performs a remarkable accuracy for fingerprint recognition. In addition, the considerable retention performance (more than 150 min, as shown in Supplementary Fig. 13) of the memristor indicates that once the offline training has been completed, the trained weights of the local memristor array can be used for recognition for a long time.

In the practical DUV fingerprint recognition task, noises are always inevitable. Here, stochastic noises with a varying scale from 1% to 20% were introduced into the input images to mimic the potential optical interference. This generates a new test set to verify the recognition capability of this in-sensor RC system based on the a-GaO$_x$ photo-synapse. Three situations, full-precision (double-precision floating-point) simulation, limited-precision (32-bit fixed-point quantization) simulation, and hardware experiment, are considered for comparison.

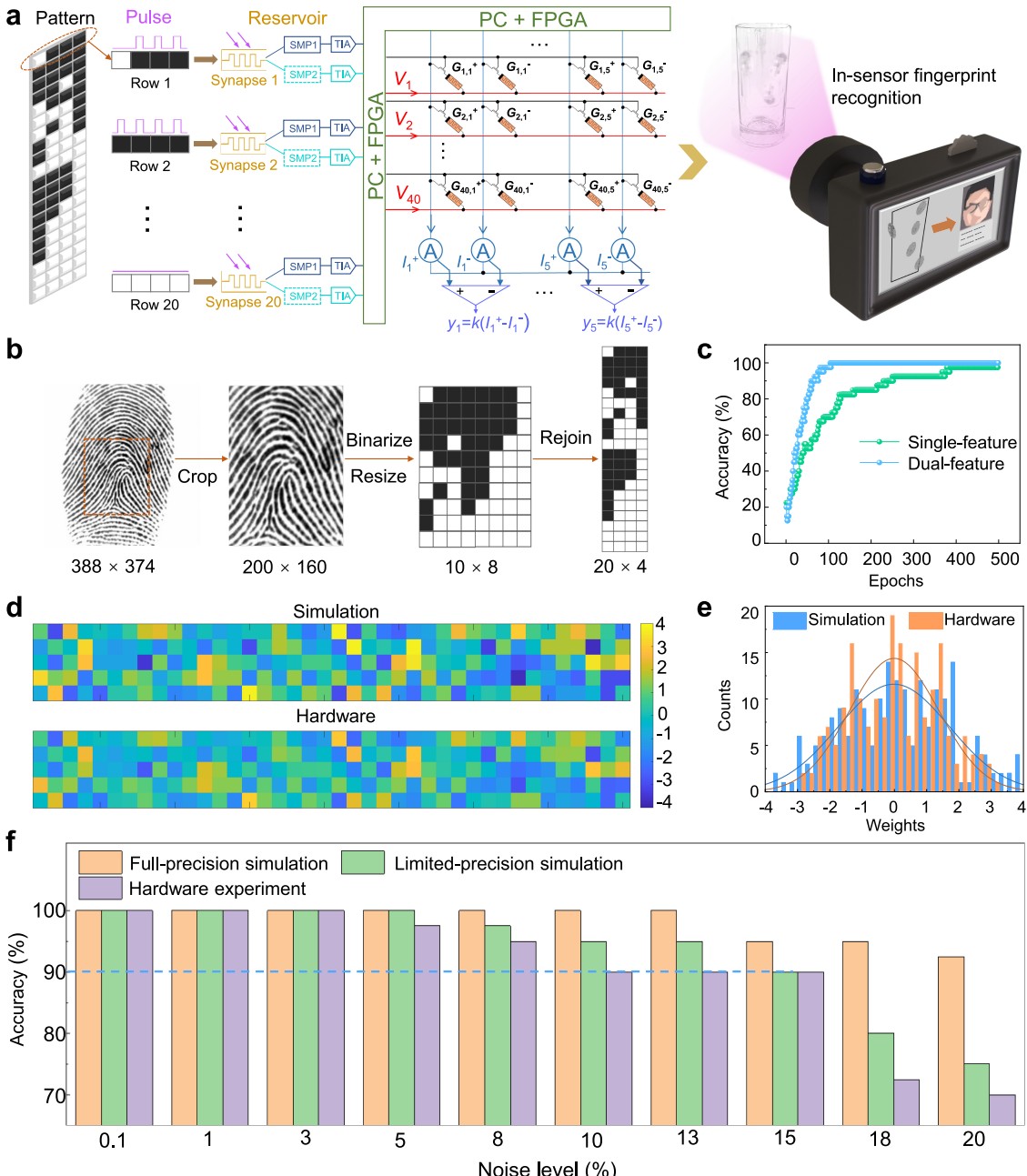

**Fig. 4 | Fingerprint recognition based on hardware DUV in-sensor RC system.** **a** Schematic of the proposed fully-hardware photoelectronic RC system for in-sensor fingerprint recognition, including photo-synapse reservoir layer which generates feature outputs, and memristor readout layer which performs network training. **b** Preprocessing method of the fingerprint images, including cropping, compressing, binarizing, and rejoining. **c** The evolution of the accuracy rates based on single and dual features during readout network training. The training process with dual features demonstrates a much faster convergence. **d** The colormaps and **e** statistic histograms of the 40 × 5 weights of the simulation and hardware experiment, respectively. The actual conductance values read from hardware were multiplied by a constant of $1.25 \times 10^4$ for better comparison with the simulated weights. **f** Influence of stochastic noise on recognition accuracy rates for fingerprint recognition of the RC system, implemented by full-precision simulation, limited-precision simulation, and hardware experiment, respectively.

As shown in Fig. 4f, recognition accuracies in all situations remain comparable under ≤3% noise level and deteriorate asynchronously with its increment. The limited resistive states of the memristor device and the amplification of non-ideal factors (*e.g.*, device-to-device and cycle-to-cycle variations, discreteness of operations, etc.) under high-level noise dominate the relatively quick deterioration in the hardware situation. Therefore, the improvement of resistive states and uniformity of the memristor devices could further improve the system robustness[52]. It is noteworthy that the recognition accuracy of the hardware experiment still maintains above 90% under 15% noise level. In summary, the fully-hardware DUV in-sensor RC system based on

a-GaO$_x$ photo-synapse has promising potential to be competent for high-precision in-situ DUV fingerprint recognition tasks.

## Discussion
In summary, we proposed a fully-hardware DUV in-sensor RC system composed of a photo-synapse reservoir layer and a memristor readout layer for latent fingerprint recognition. It is found that the oxygen-vacancy-related hole traps dominate the PPC effect and induce the nonlinear neuromorphic features of the a-GaO$_x$ DUV photo-synapse for in-senor RC. As a result, the inputs of the reservoir can be non-linearly mapped to dimensionality-reduced outputs, which constitute

the feature space. Acting as the readout network, memristor device array with analog conductivity takes charge of the training of reservoir outputs and parallel in-memory computing. Based on such a hardware system, the high recognition accuracy of DUV fingerprint images nearly matches the simulation results when adopting a dual-feature strategy. The system achieves 100% recognition accuracy after 100 training epochs and maintains 90% accuracy even under 15% background noise level consistent with the anti-noise characteristics of DUV light. This fully-hardware DUV in-sensor RC system provides a prototype for efficient identification and security applications.

## Methods

### Device fabrication

After the first lithography process, the 208 nm a-GaO$_x$ thin film was deposited onto the low-resistance p-type Si substrate with 300 nm SiO$_2$ by RF-magnetron sputtering at room temperature. Samples S1, S2, and S3 were fabricated by sequentially increasing oxygen content. Separated a-GaO$_x$ films were obtained after the lift-off process. The rapid thermal annealing process of the a-GaO$_x$ thin films was performed at 400 °C in ambient N$_2$ for 1 min. After another lithography process, the source and drain electrodes consisting of Ti/Au (20/50 nm) were deposited onto the a-GaO$_x$ thin film by electron beam evaporation.

### Film characterizations

The relative contents of $O_I$ and $O_{II}$ in the a-GaO$_x$ film were determined by X-ray photoelectron spectroscopy (XPS, ESCALAB 250Xi). The morphology of the a-GaO$_x$ film was characterized by atomic force microscope (AFM, Bruker Dimension Icon). The potential distributions of the samples were obtained by above AFM instrument under KPFM mode. In the target regions ($5 \times 5\,\mu m^2$), only the middle parts ($1 \times 5\,\mu m^2$) were injected. After the carrier injection, the surface potential of the target region was scanned every 9 min.

### Device characterizations

The electronic and photoelectronic characteristics, including $I$–$t$ photoresponse curves, PPF effects, and reservoir feature outputs of the a-GaO$_x$ photo-synapse, were measured at room temperature using an Agilent B1500A Semiconductor Device Analyzer. An LED with a wavelength of 254 nm was used as the monochromatic light source, and the light intensity was calibrated by an optical power meter (S401C and PM100D). A shutter (Thorlab SHB-025T) was used to modulate the pulse waveform of the DUV light source. During all the measurements, the bias was fixed at 1 V. All experiments for RC were performed under the optimal pulse condition of 25 ms width and 45 ms interval.

### Statistics and reproducibility

Experiments were reproducible. All the error box figures depicted by Origin software are unified with the same standard: box with percentile range from 25 to 75, whisker with outlier coefficient of 1.5, median with solid line and mean with circle dot.

### Memristor array fabrication

The above-mentioned 1K-bit array adopts the 0.18 μm standard technology of Semiconductor Manufacturing International Corporation (SMIC). The memristor composed of TiN/TaO$_x$/HfO$_y$/TiN was stacked by the following steps: First, the TiN layer was deposited by physical vapor deposition as the bottom electrode. Next, the HfO$_y$ and TaO$_x$ layers were successively stacked on the TiN layer by atomic layer deposition. Finally, another TiN layer was deposited as the top electrode using the same process as the bottom layer.

### Basic memristor array operations

The current values of the reservoirs are transmitted to transimpedance amplifiers (TIAs) to convert them into voltage values,

which are then fed into the memristor array. Each differential pair in the memristor array represents a single weight of the neural network. Transistors are used for device addressing and crosstalk current suppression. As for the training of the memristor array, we utilized an offline training method to update the weight (conductance) matrix of the array. Once the software simulation is completed, the weights of the whole array (400 memristor devices) are updated by referring to the simulation results, column by column. To SET a selected column, all source lines (in blue, in Fig. 4a) were floated, except the selected one, which was grounded. All word lines (in red) were biased at the same SET voltages. Each bit line (in black) was assigned with a different voltage based on the targeted conductance of the selected memristor. For RESET operation, word lines of selected columns were grounded, all source lines were biased with RESET voltages, and each bit line was biased with a large voltage to allow sufficient RESET current.

### Network training

All the reservoir outputs were used to construct the input layer of a fully-connected network: for the single-feature strategy, the network size was $20 \times 5$; for the dual-feature strategy, the network size was $40 \times 5$. The fully-connected network was trained by the MATLAB Deep-learning Toolbox, utilizing the Softmax output function and the logistic regression to supervise the learning. The stochastic noise was made by the dot product of the MATLAB randn matrix and the grayscale value throughout the whole image. The final noise was obtained by the product of the noise level and the stochastic noise and then added into the original images.

## Data availability

All data needed to evaluate the conclusions in the paper are presented in the paper and/or the Supplementary Information. The data that support the plots within the paper and other findings of this study are available from the corresponding authors upon reasonable request.

## Code availability

All code used in this study is based on MATLAB scripts and toolboxes, and available from the corresponding authors upon reasonable request.

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

## Acknowledgements

This work was supported by the National Natural Science Foundation of China under Grant nos. 61925110 (S.L.), U20A20207 (S.L.), 62004184 (G.X.), 62004186 (X.L.Z.), 62104044 (X.M. Z.), 61825404 (Q.L.) and 51961145110 (S.L.), the Key Research Program of Frontier Sciences of Chinese Academy of Sciences under Grant No. QYZDB-SSW-JSC048 (S.L.), the Key-Area Research and Development Program of Guangdong Province under Grant No. 2020B010174002 (S.L.), the China Postdoctoral Science Foundation under Grant Nos. 2020M671895 (X.L.Z.) and BX20200320 (X.L.Z.), and the Opening Project of the Key Laboratory of Microelectronics Devices and Integration Technology in Institute of Microelectronics of CAS and the Key Laboratory of Nanodevices and Applications in Suzhou Institute of Nano-Tech and Nano-Bionics of CAS. This work was partially carried out at the Center for Micro and Nanoscale Research and Fabrication of USTC.

## Author contributions

X.L.Z., X.M.Z., and S.L. conceptualized the system and related application. Z.Z., X.H., and X.M. fabricated the photo-synapses. Z.Z., S.T., and Y.Z. characterized the photo-synapses and memristor array. Z.Z. and

X.L.Z. wrote the original manuscript. Z.Z., X.L.Z., X.M.Z., G.X., Q.L., and S.L. participated in the manuscript writing, reviewing, and editing.

## Competing interests

The authors declare no competing interests.
