## [Peer Review File · Nature Communications]

REVIEWER COMMENTS

Reviewer #1 (Remarks to the Author):

In this paper, the authors introduced a deep ultraviolet (DUV) in-sensor reservoir computing (RC) system to perform in-situ latent fingerprint recognition. The proposed system features compactness and high-power efficiency achieved via implementing amorphous Gallium-oxide sensors with enhanced persistent photoconductivity effect to emulate the photo-synapse reservoir layer, whereas the output layer is implemented using HfOx/TaOx based memristive array. During verification, the proposed system is trained and tested with fingerprint images. With that, 90% recognition accuracy is achieved even in the presence of 15% noise.

Suggestions:

- Since there are multiple variants of reservoir networks, it would be useful to include the mathematical model of the reservoir network topology.
- The reservoir networks are known to have lateral connections [feedback] to enable the representation of temporal context. Did the authors use lateral connections? if the answer is no, this cannot be a reservoir network.
- Using the term in-situ implies that the fingerprint recognition, including preprocessing, scaling, etc, is performed on-site, but this is not the case here. All the test images are preprocessed off-site. The same is applied to RC system training.
- In Fig.4-f, is there any explanation for equal classification accuracies of the software and hardware models when the noise level is below 3%? Why is it not the case for higher noise levels?
- I would recommend verifying the proposed RC system with unseen fingerprint images rather than using noisy training images as a test set.
- In the hardware model, the input features are re-represented to expand in the temporal domain. Thus, it is unclear how did the authors employ the conventional backpropagation to train the network.
- The authors mentioned that the proposed system is power efficient as compared to ex-situ latent fingerprint recognition system. Are there any quantitative results to support this claim?
- Given the fact that memristor conductance may change over time, how often do we need to re-train the memristive array of the readout layer to maintain consistent performance?
- In Line 390, it is mentioned “current values of the reservoirs are transmitted to trans-impedance amplifier to convert them into voltage values.” Trans-impedance amplifier converts voltage to current! Thus, the amplifier name should be replaced by trans-resistance amplifier.

- There are a few grammatical mistakes need to be fixed.

Reviewer #2 (Remarks to the Author):

The authors proposed an in-sensor reservoir computing system for in-situ latent fingerprint recognition. In such a system, GaOX photodetector acts as the deep ultraviolet photo-synapses for information input and the memristor array is utilized as the training and readout layer. Systematic experiments have been performed, including the engineering of GaOX component to improve the photo-synapse behavior, mapping of complex input vectors into dimensionality-reduced output vectors, and configuring and simulating of the whole in-sensor reservoir computing system. The authors demonstrate the nonlinear mapping characterization of input and output based on the GaOX photoelectric reservoir and proposed dual-feature strategy for feature sharpening. Especially, this hardware system maintains high accuracy above 90% for fingerprint recognition even under 15% background noise level. This prototype system for image recognition combining photo-synapses and memristors will provide more insight into emerging in-sensor reservoir computing. Overall, the topic of this work is truly interesting. The manuscript is well organized. I would recommend the acceptance if the authors can address below questions.

1. The authors modulate the PPC effect with a longer decay process by decreasing the O contents unilaterally. The authors are suggested to clarify the factors that determine the PPC effect. In addition, please make it clear in the main text, what are the detailed requirements in synapse behavior for in-sensor reservoir computing?
2. The authors mentioned that “the deliberately enlarged PPC effect by Ga-rich design turns the sample S1 into an ideal photo-synapse”. But there must be something wrong in Fig. 2, where the main information about S1, S2, and S3 are missing. Even the main text and caption introduce the figures in details, Fig. 2 and Supplementary Fig. 2 have been mistakenly labelled.
3. The trends during input mapping in Supplementary Fig. 6 and Fig. 7 are similar. How much will the difference in peak value influence the recognition accuracy?
4. The dual-feature strategy sharpens the feature of various inputs and improves the recognition accuracy. But it also increases the burden of the readout layer. Can the authors comment this effect on the overall performance?
5. In Supplementary Fig. 11, pulse stimulations for increment and decrement of memristor conductance are missing. Also, the description of the training method of the memristor array is unclear in method section. The authors should make it more clear.

6. “differentcomplicance” in Supplementary Fig. 9 should be “different compliance”. Please check the English throughout the manuscript.

**Response to Reviewer's Comments**

This Response Letter is regarding a former manuscript submitted to *Nature*
*Communications*, entitled “*In-sensor reservoir computing system for latent*
*fingerprint recognition with deep ultraviolet photo-synapses and memristor array*”
by Zhongfang Zhang et al. (NCOMMS-22-15956). We would like to express our
special thanks to the reviewers, for their useful comments have guided us to improve
the manuscript quality effectively. We revised both the main text and the supplementary
information (SI) accordingly, and the detailed responses to the questions and comments
are summarized. In the following point-to-point response, the original comments are in
black fonts, and our responses are in blue fonts. Changes in the revised main text and
SI are highlighted in yellow.

14 **I. Comments from Reviewer 1**

**Overall Comment:**

In this paper, the authors introduced a deep ultraviolet (DUV) in-sensor reservoir
computing (RC) system to perform in-situ latent fingerprint recognition. The proposed
system features compactness and high-power efficiency achieved via implementing
amorphous Gallium-oxide sensors with enhanced persistent photoconductivity effect to
emulate the photo-synapse reservoir layer, whereas the output layer is implemented
using $\text{HfO}_x/\text{TaO}_x$ based memristive array. During verification, the proposed system is
trained and tested with fingerprint images. With that, 90% recognition accuracy is
achieved even in the presence of 15% noise.

**Reply to Overall Comment:** We thank the referee for the precious time and
constructive comments on our manuscript. Our responses to the comments one by one
are shown as follows.

**Comment 1:** Since there are multiple variants of reservoir networks, it would be useful
to include the mathematical model of the reservoir network topology.

**Reply to Comment 1:** We thank the reviewer for this suggestive comment.

The echo state network (ESN) is a fitted model for understanding a general RC
architecture, as shown in Fig. R1a. The I/O relationships can be represented by the

formulas:

$$34 \quad x(t + 1) = f(W_{res} \cdot x(t) + W_{in} \cdot u(t))$$

$$35 \quad y(t) = W_{out} \cdot x(t)$$

Among ESN models, the delayed-feedback system is mostly suitable for temporal
 information classification. The excitations of the physical node in response to the
 delayed signals can effectively act as a chain of virtual nodes, as shown in Fig. R1b.
 The temporal transformation of the reservoir state can be represented by the formula:

$$40 \quad dx(t)/dt = F(t, x(t), x(t - \tau))$$

$$41 \quad \theta = \tau/N$$

where F is the system function determined by intrinsic material properties, τ is the
 duration time, N is the number of nodes, and θ is the time-step. According to the above
 characteristics, we can conclude that the performance of the device used for RC requires
 two features: the nonlinear response to continuous input and the short-term decay
 characteristic, which are also theoretically explained in other works^{R1, 2}.

**Fig. R1 Schematic diagram of the mathematical model of the typical RC topology.**

**a** General echo state network model. **b** Delayed-feedback RC model.

In order to further confirm that the device meets the above two characteristics,
according to the suggestions of the reviewer, we further analyzed the nonlinear response
curves of the device, as shown in Fig. R2. The response formula under continuous light
input and the decay formula after light input of the device have been analyzed:

i) The response function can be represented by:

$$56 \quad R = R_0 + A [1 - \exp(-(t-t_0)/\tau)]$$

$$57 \quad A = f(R_0)$$

$$58 \quad \tau = g(R_0)$$

where R_0 is the initial current of the response process, A is the difference between $R(\infty)$
and R_0 , and t_0 is the starting time of the response process. The fitting parameters A and
τ are related to the initial current state R_0 , and the functions f and g are both determined
by the intrinsic characteristics of the device and the input light power.

Thus, the fitting result of the example in Fig. R2 is extracted to be:

$$64 \quad R = 4 + 39 [1 - \exp(-(t-t_0)/108)]$$

where R is in nA and $t-t_0$ is in ms.

ii) The decay functions can be represented by:

$$67 \quad D = D_0 - A' [1 - \exp(-(t-t_0)/\tau')]$$

$$68 \quad A' = r(D_0)$$

$$69 \quad \tau' = s(D_0)$$

where D_0 is the initial current of the decay process, A' is the difference between D_0 and
$D(\infty)$, and t_0 is the starting time of the decay process. The fitting parameters A' and τ'
are related to the initial current state D_0 , and the functions r and s are both determined
by the intrinsic characteristics of the device.

Thus, the fitting result of the example in Fig. R2 is extracted to be:

$$75 \quad D = 26 - 10 [1 - \exp(-(t-t_0)/47)]$$

where D is in nA and $t-t_0$ is in ms.

From these two processes, we can deduce that the characteristics of the device can meet
the requirements for RC.

**Fig. R2 Nonlinear functions (current versus time) extracted from the**
 **photoresponse curves of the device.** The response function under continuous light
 input is represented by R and the decay function is represented by D .

In this work, referring to the time-delayed reservoir model, we built a parallel
 architecture of the time-delayed reservoir. As shown in Fig. R3, the dots in yellow
 actually represent the virtual nodes, which are set at the pulse ending edge with a fixed
 interval θ . For the designed readout, only the last 1 or 2 nodes are utilized to construct
 the output vector. Multiple reservoirs in parallel are utilized to accomplish one
 comprehensive output from a binary image. As for the RC network training, only the
 readout matrix (W_{out}) needs to be trained, and all the other connections are fixed^{R3,4}.

**Fig. R3 Schematic diagram of the parallel time-delayed reservoir network as a**
 **demonstration of our work.** The image is divided suitably then inputted into the
 reservoirs in parallel. The virtual nodes of each reservoir are coupled with a time
 interval θ . For the designed readout network, only the last 1 or 2 nodes of each reservoir
 are utilized to construct the output vector.

According to this comment, we have supplemented the relevant figures and
 demonstrated the mathematical model:

(main text, **Fingerprint recognition with fully-hardware DUV in-sensor RC system,**
 **Paragraph 1)**

“To verify the feasibility of DUV in-sensor RC for fingerprint recognition, we
 constructed a hardware system, composed of a photo-synapse reservoir layer and a
 memristor readout layer, as shown in Fig. 4a. The relationship between the
 mathematical model and the physical hardware of this system has been illustrated (see
 Supplementary Fig. 8). In such a system for DUV fingerprint recognition, the images
 are first converted into DUV light pulses.”

(SI, **Supplementary Fig. 8)**

**Supplementary Fig. 8 Schematic diagram of the parallel time-delayed reservoir**
 **network as a demonstration of our work.** The image is divided suitably then input
 into the reservoirs in parallel. The virtual nodes of each reservoir are coupled with a
 time interval θ . For the designed readout network, only the last 1 or 2 nodes of each
 reservoir are utilized to construct the output vector.”

**Comment 2:** The reservoir networks are known to have lateral connections [feedback]
 to enable the representation of temporal context. Did the authors use lateral connections?
 if the answer is no, this cannot be a reservoir network.

**Reply to Comment 2:** We thank the reviewer for this suggestive comment.

The schematic of the lateral connections can be reflected in our reservoir model (Fig.
 R3), since the state change of the reservoir is not only related to external input, but also
 related to the real-time state of the reservoir (conductivity of the device). Despite the
 independence of multiple reservoirs, the lateral connections do exist between the virtual
 nodes of one reservoir, which can be reflected by the time-dependent pulse responses
 of I-t curves. As an example, although the last pulse is the same “1”, the sampling

currents (SMP1) of “1101”, “0011” and “0001” are different, as shown in Fig. R4. If
there is no lateral connection (feedback), the current obtained will only be related to the
last input (namely “1”) in the 4-bit sequences.

**Fig. R4** A typical example of lateral connection by different 4-bit pulse inputs,
including “1101”, “0011”, and “0001”. Although the last pulse is “1”, the
characteristic currents (SMP1) of “1101”, “0011” and “0001” are different. The state
change of the reservoir is not only related to external input, but also related to the real-
time state of the reservoir.

According to this comment, we have supplemented correlated sentences in the main
text (**Nonlinear mapping of 4-bit inputs of the a-GaO_x DUV reservoir, Paragraph**
**2**): “To illustrate the feature sampling, the I-t curves of three representative inputs of
“0001” (in blue), “0011” (in red), and “1101” (in purple) of the a-GaO_x reservoir are
exhibited in Fig. 3b. Although the last pulses are all “1”, their decay processes after the
input sequences are different. Therefore, the final state of the reservoir not only relates
to the last input, but also depends on its real-time state, indicating the lateral
connections in such an a-GaO_x reservoir^{21,22}. Based on the conspicuous difference, each
pixel sequence can be featured by current sampling to realize feature extraction”.

**Comment 3:** Using the term in-situ implies that the fingerprint recognition, including

preprocessing, scaling, *etc.*, is performed on-site, but this is not the case here. All the
test images are preprocessed off-site. The same is applied to RC system training.

**Reply to Comment 3:** We thank the reviewer for this helpful comment. It is reasonable
to say that the image preprocessing is obviously not in situ. The reason for our
preprocessing is that the off-site is limited by the size of the memristor array. Therefore,
we reasonably believe that this work verifies the prototype of a fingerprint recognition
system and provides potential inspirations for the realization of in-situ fingerprint
recognition system.

According to the reviewer's comment, we have corrected the descriptions about "in-
situ" in the main text and SI, especially the title of this work.

The revised title is "**In-sensor reservoir computing system for latent fingerprint**
**recognition with deep ultraviolet photo-synapses and memristor array**".

Other revisions about "in-situ" in the revised main text and SI are all highlighted in
yellow.

**Comment 4:** In Fig.4-f, is there any explanation for equal classification accuracies of
the software and hardware models when the noise level is below 3%? Why is it not the
case for higher noise levels?

**Reply to Comment 4:** We thank the reviewer for this helpful comment.

The fingerprint images with increasing noise levels are exhibited with different
resolutions, as shown in Fig. R5, in which the binary images are the most crucial. While
the noise level is below 3%, the noise is negligible and the difference between the
training set and the test set is not obvious. This as-trained weight matrix can be
competent for the recognition task through both the simulation and the hardware,
leading to the relatively close classification accuracies.

Most pixels in the binary image have changed at above 3% noise level, visible to the
naked eye. The weight matrix of software simulation is based on double-floating
number, while the precision of hardware (memristor) is limited by the quantity of the
controllable conductance states. In addition, there are various device non-idealities of
memristor hardware (*e.g.*, device-to-device and cycle-to-cycle variations, discreteness

of operations, *etc.*), which will lead to the attenuation of accuracies with the
 introduction of noises. According to the above factors, the gap of accuracy between the
 software and hardware model becomes increasingly obvious with the increase of noise
 level.

In summary, the main factors resulting in the differences in classification accuracies
 could be as follows: i) the images exhibit apparent changes when the noise level is
 above 3%. ii) the limited conductance states of the memristor hardware affect the
 calculation precision of the readout layer. Similarly, in the reports of Midya. R. *et. al.*^{R5},
 the recognition accuracy of hardware verification also faces attenuation while
 increasing the noise.

 **Fig. R5 Representative fingerprint images with increasing noise levels displayed**
 **in different pixel sizes.** While the noise level is below 3%, the changes of the binary
 image only take place in a few pixels; however, when the noise level is above 3%, the
 features of most pixels in the image have changed obviously, even visible to the naked
 eye.

According to this comment, we have supplemented the correlated sentences in the main
 text (**Fingerprint recognition with fully-hardware DUV in-sensor RC system,**
 **Paragraph 3**):

“Three situations, full-precision (double-precision floating-point) simulation, limited-

precision (32-bit fixed-point quantization) simulation, and hardware experiment, are
considered for comparison. As shown in Fig. 4f, recognition accuracies in all situations
remain comparable under $\leq 3\%$ noise level and deteriorate asynchronously with its
increment. The limited resistive states of the memristor device and the amplification of
non-ideal factors (*e.g.*, device-to-device and cycle-to-cycle variations, discreteness of
operations, etc.) under high-level noise dominate the relatively quick deterioration in
the hardware situation. Therefore, the improvement of resistive states and uniformity
of the memristor devices could further improve the system robustness⁵¹. It is
noteworthy that the recognition accuracy of the hardware experiment still maintains
above 90% under 15% noise level. In summary, the fully-hardware DUV in-sensor RC
system based on a-GaO_x photo-synapse has promising potential to be competent for
high-precision in-situ DUV fingerprint recognition tasks.”

**Comment 5:** I would recommend verifying the proposed RC system with unseen
fingerprint images rather than using noisy training images as a test set.

**Reply to Comment 5:** We thank the reviewer for this helpful comment. The noises for
practical recognition scenes inspired us to perform such a comparison in the original
manuscript. According to the reviewer’s comment, we also supplemented the
simulation experiment with the proposed method.

Unlike the large MNIST handwriting database, this fingerprint database (Fingerprint
Verification Competition 2002 database) has relatively small sample size. There are
only 8 fingerprint images for each person in the original data set, which is limited for
the division of the training and test sets. Thus, we conducted a simple extension of the
data set by introducing one random noise pixel in the binary image for 10 times, thus,
there are 80 available images for each person, with a total of 400 images. Then, we
divide the extended fingerprint images into the 80% training set and the 20% test set
(namely the unseen images), as shown in Fig. R6a. By utilizing the dual-feature strategy
of the reservoir, the simulated recognition accuracy for trained fingerprint images is
beyond 96% after 1000 training epochs, as shown in Fig. R6b. As for the test set, the
confusion matrix is shown in Fig. R6c, indicating an excellent recognition accuracy of
92.5% for the recognition of the unseen images.

**Fig. R6 Recognition simulation of the unseen fingerprint images.** **a** Expansion of
 the data set of the fingerprints from 40 to 400 images by introducing one random noise
 pixel in each binary image for 10 times (taking the C-1 image in Supplementary Fig.
 10 as an example), owing to the finite scale of the FVC 2002 database. 80% of the
 fingerprint images were set as the training set and the other 20% as the test set (namely
 the unseen images). **b** Accuracy convergency during the training process within 1000
 epochs. Considerable recognition accuracy can be achieved upon certain training
 epochs. **c** Confusion matrix of the fingerprint recognition with the unseen images as the
 test set. The test accuracy is extracted to be 92.5%.

According to this comment, we have supplemented the recognition simulation of
 untrained fingerprint images and added correlated contents in both main text and
 Supplementary Information:

(main text, **Fingerprint recognition with fully-hardware DUV in-sensor RC system,**
 **Paragraph 1)**

“Thus, a dual-feature strategy is employed, and only a dimensionality-reduced 40×5
 weight matrix needs to be trained for each fingerprint image. As an example, a
 recognition accuracy for unseen fingerprint images has been simulated to be around 92%
 based on this dual-feature strategy, where the expanded sample amounts of the

fingerprint images ensure the high recognition accuracy (see Supplementary Fig. 11).”

(SI, Supplementary Fig. 11)

“

**Supplementary Fig. 11 Recognition simulation of the unseen fingerprint images. a**

Expansion of the data set of the fingerprints from 40 to 400 images by introducing one

random noise pixel in each original image for 10 times (taking the C-1 image in

Supplementary Fig. 10 as an example), owing to the finite scale of the FVC 2002

database. 80% of the fingerprint images were set as the training set and the other 20%

as the test set (namely the unseen images). **b** Accuracy convergency of the training

process within 1000 epochs. Considerable recognition accuracy can be achieved upon

certain training epochs. **c** Confusion matrix of the fingerprint recognition with the

unseen images as the test set. The test accuracy is extracted to be 92.5%.”

**Comment 6:** In the hardware model, the input features are re-represented to expand in

the temporal domain. Thus, it is unclear how did the authors employ the conventional

backpropagation to train the network.

**Reply to Comment 6:** We thank the reviewer for this comment. The backpropagation

is usually applied in multilayer neural networks, updating the weights from the output

layer to the input layer. In our work, we only train the single-layer readout network and
select the Softmax as the output function, then the readout weights are updated by
logistic regression to minimize the loss. Unlike the traditional method in multilayer
neural networks, the backpropagation algorithm is not used in the training process,
since the weights in the reservoir are always fixed.

According to this comment, we have corrected the descriptions of the readout network
training methods (**Methods, Network training**):

“The fully-connected network was trained by the MATLAB Deep-learning Toolbox,
utilizing the Softmax output function and the logistic regression to supervise the
learning. The stochastic noise was made by the product of the MATLAB randn matrix
and the grayscale value throughout the whole image.”

**Comment 7:** The authors mentioned that the proposed system is power efficient as
compared to ex-situ latent fingerprint recognition system. Are there any quantitative
results to support this claim?

**Reply to Comment 7:** We thank the reviewer for this helpful comment.

The energy consumption of our system includes optoelectronic reservoirs and
memristor array. From the perspective of quantitative calculation, the power
consumption of the optoelectronic reservoir can be extracted by the formula $E=IVt$. In
our work, by setting 20 nA as the average current of the reservoir state (see Fig. 3d in
the main text), the energy consumption per pulse operation of the reservoir is calculated
to be $E=20 \text{ nA} \times 1 \text{ V} \times 25 \text{ ms}=0.5 \text{ nJ}$. Using the same calculation method, the similar
optoelectronic synapse in previous report costs approximately 85 nJ per operation of
optical information processing^{R6}, indicating that the optoelectronic reservoir in our
work is much more energy-efficient than the former report. In addition, this pulse
operation of the reservoir contains both the sensing and processing of the optical
temporal information. The traditional systems require sensors and photoelectric signal
converters, while the increased energy consumption of these additional parts is usually
not mentioned in the reports to conduct quantitative calculation^{R3, 7}.

As for the training consumption, taking the SET operation of one memristor device as
an example (see Supplementary Fig. 12), the power consumption can be approximately
extracted by $E'=gV^2t=300 \text{ } \mu\text{S} \times (2.5 \text{ V})^2 \times 500 \text{ } \mu\text{s}=0.938 \text{ } \mu\text{J}$. Actually, we have

introduced the dimensionality-reduced conception in our work, which means the
introduction of the reservoir architecture will reduce the scale of memristor array for
the readout training. Taking the 10×8 image in our work as an example, if there is no
in-sensor reservoir, the readout network requires 800 (80×5×2) memristor devices. By
utilizing the dual-feature strategy of 20 reservoirs, the amounts of memristor will
decline by half. Since the energy consumption of a memristor is approximately 1000
307 times larger than that of a reservoir, we can deduce that the reduction of dimensionality
is valuable for the overall energy-efficiency. Therefore, from the quantitative
calculation, the reservoir architecture in our work possesses potential energy-efficient
characteristic.

According to this comment, we have updated the demonstrations of the energy
consumption of the reservoir (**Nonlinear mapping of 4-bit inputs of the a-GaO_x DUV
reservoir, Paragraph 3**):

“Consequently, the feature space based on nonlinear photoresponse configures the
classification process of the reservoir, reducing the dimensionality of raw data from 4-
bit digital inputs to 2 analog outputs that serve as the inputs of the linear readout layer⁴⁹,
⁵⁰. The energy consumption per pulse operation of the optoelectronic reservoir can be
estimated to be $E=20\text{ nA}\times 1\text{ V}\times 25\text{ ms}=0.5\text{ nJ}$, indicating that the reservoir architecture
possesses potential energy-efficient characteristic.”

**Comment 8:** Given the fact that memristor conductance may change over time, how
often do we need to re-train the memristive array of the readout layer to maintain
consistent performance?

**Reply to Comment 8:** We thank the reviewer for this comment. Considering the
resistance decay of the memristor, the retention characteristic measurement of our
memristor is conducted by 150 minutes, as shown in Supplementary Fig. 13. Therefore,
within the retention time, re-train is not necessary. Namely, the re-train time is greater
than 150 minutes, which is comparable to the previous reports^{R8}, and sufficient for an
identification system.

**Comment 9:** In Line 390, it is mentioned “current values of the reservoirs are
transmitted to trans-impedance amplifier to convert them into voltage values.” Trans-
impedance amplifier converts voltage to current! Thus, the amplifier name should be
replaced by trans-resistance amplifier.

**Reply to Comment 9:** We thank the reviewer for this comment. Maybe there are some
misunderstandings in the English expression of the circuit element, since the nouns
“impedance” and “resistance” represent the similar physical quantity in ohm (Ω). A
trans-impedance amplifier is usually utilized to convert the current signals to voltage
signals^{R5,9}, as shown in Fig. R7. When the resistance R is fixed, V_{out} could be a simple
multiplication of the analog current of reservoir I_i and the constant R , implementing the
function of converting the current value into a voltage value.

**Fig. R7 Schematic diagram of a trans-impedance amplifier (TIA) model in our**
**work.** In this work, the TIA elements convert the current outputs of the reservoirs into
voltage values, namely $V_{out}=I_iR$.

**Comment 10:** There are a few grammatical mistakes need to be fixed.

**Reply to Comment 10:** We thank the reviewer for this helpful comment. According
to the reviewer’s comments. The English expression of the full text has been checked
and polished. All the revisions about typos and grammar in the revised main text and
SI are highlighted in yellow.

**II. Comments from Reviewer 2**

**Overall Comment:**

The authors proposed an in-sensor reservoir computing system for in-situ latent
fingerprint recognition. In such a system, GaO_x photodetector acts as the deep
ultraviolet photo-synapses for information input and the memristor array is utilized as
the training and readout layer. Systematic experiments have been performed, including
the engineering of GaO_x component to improve the photo-synapse behavior, mapping
of complex input vectors into dimensionality-reduced output vectors, and configuring
and simulating of the whole in-sensor reservoir computing system. The authors
demonstrate the nonlinear mapping characterization of input and output based on the
GaO_x photoelectric reservoir and proposed dual-feature strategy for feature sharpening.
Especially, this hardware system maintains high accuracy above 90% for fingerprint
recognition even under 15% background noise level. This prototype system for image
recognition combing photo-synapses and memristors will provide more insight into
emerging in-sensor reservoir computing. Overall, the topic of this work is truly
interesting. The manuscript is well organized. I would recommend the acceptance if the
authors can address below questions.

**Reply to Overall Comment:** We thank the referee for the positive comments on the
significance of our work. Our responses to the comments one by one are shown as
follows.

**Comment 1:** The authors modulate the PPC effect with a longer decay process by
decreasing the O contents unilaterally. The authors are suggested to clarify the factors
that determine the PPC effect. In addition, please make it clear in the main text, what
are the detailed requirements in synapse behavior for in-sensor reservoir computing?

**Reply to Comment 1:** We thank the reviewer for this helpful comment.

There are several factors to introduce PPC effects in semiconductor materials, such as
ionization of oxygen vacancy sites^{R10}, macroscopic potential barriers^{R11}, and metastable
peroxides^{R12}. In the previous report of photoelectronic device based on amorphous
Ga₂O₃^{R13}, the oxygen vacancy is a relatively crucial factor to cause the PPC effect.

Researchers have reported many methods to modulate the PPC effect, including oxygen
ambient modulation^{R14}, post annealing^{R15}, and Ar-plasma pretreatment^{R16}. By utilizing
these methods in the process of material growth, the PPC effect can be well controlled,

whether it is enhanced or vanished. In this work, we fabricated comparative samples of
various O contents by ambient modulation, and validated the influence of oxygen
vacancy on the PPC effect.

In addition, we have summed up some requirements for photo-synapse to be used in
reservoir computing^{R2, 7}:

a) Nonlinearity

The nonlinearity in the RC is mainly shown in the nonlinearity of the neurons. This
setup enables the RC to cope with the nonlinear functions in real world. There have
also been reports using a nonlinear dynamical system in the state updating of the RC,
reaching good result in time-series processing. As for the photo-synapse, amorphous-
Ga₂O₃-based device have inherent nonlinear photoresponse (see Supplementary Fig. 2),
thus are candidates for an implementing physical RC.

b) Short-term memory

The short-term memory is a component of the echo state property, the condition for the
reservoir to reach an asymptotic stability that the states of the reservoir network is
determined by the input and the real-time reservoir state, thus the reservoir can show
good performance in tracking and synchronizing with a time series. In the situation of
the photo-synapses, we would require the devices to show decay in the photogenerated
conductance after illumination. Interestingly, the PPC effect which represents the decay
process of the photogenerated current, could be regarded as the STM characteristic of
a synaptic device.

c) High dimensions/More reservoir states

The function of RC largely relies on the ability of dimension upgrading. In the
dimension upgrading process, the input is mapped into a space of higher dimension,
and linear separation is done to give prediction of the time series data points. As for
optoelectronic reservoir, it usually requires that the photo-synapse can generate more
states when given with any type of input optical data.

**Fig. R8 Gradual state change by conducting consecutive pulse stimulations.** With
 the increasing of pulse numbers, the conductance of the reservoir rises nonlinearly,
 indicating abundant reservoir states.

419 d) Stability

The stability (endurance) is an important property in implementing the RCs. It requires
 that the reservoir could maintain its original properties like the decay constant, upper
 and lower limits of the conductance, and so on. It is hoped that the hardware platform
 can be effective and also endurable, since the RC system must be trained before they
 are introduced in real-world applications.

**Fig. R9 Repeatability of one typical device (Device #7 in Supplementary Fig. 7) as**
 **a demonstration of endurance performance.** Each box includes 100 operations of the
 same pulse inputs.

According to the reviewer's comments, we have claimed the detailed features in
 synapse behavior for in-sensor reservoir computing in the revised main text

**(Introduction, Paragraph 2):**

“Fortunately, a promising strategy of in-sensor **RC based** on optoelectronic devices has
been proposed for temporal sensory information processing and verified with the
assistance of system simulation^{21, 22}. **In order to fulfill the in-sensor applications, the**
**optoelectronic devices should be marked by the properties of nonlinearity response,**
**short-term memory (STM), multiple states and stability.** Nevertheless, the waveband
utilized in **above** works is not suitable for DUV detection.”

**Comment 2:** The authors mentioned that “the deliberately enlarged PPC effect by Ga-
rich design turns the sample S1 into an ideal photo-synapse”. But there must be
something wrong in Fig. 2, where the main information about S1, S2, and S3 are
missing. Even the main text and caption introduce the figures in details, Fig. 2 and
Supplementary Fig. 2 have been mistakenly labelled.

**Reply to Comment 2:** We thank the reviewer for this helpful comment. Really sorry
about the faults for Fig. 2, and Supplementary Fig. 2. We have modified the relevant
figures and captions in the revised manuscript as:

(main text, **Fig. 2**)

“

**Fig. 2 PPC effect and synaptic behavior of the a-GaO_x DUV sensor.”**

(SI, Supplementary Fig. 2)

“

**Supplementary Fig. 2 a-GaO_x device and its nonlinear photoresponse. a** Schematic
 diagram of the cross-section structure of the a-GaO_x device. **b** Nonlinear dependence
 of ΔI on DUV light pulse width (25 ms) under various power densities.”

**Comment 3:** The trends during input mapping in Supplementary Fig. 6 and Fig. 7 are
 similar. How much will the difference in peak value influence the recognition accuracy?

**Reply to Comment 3:** We thank the reviewer for this helpful comment. It is found
that the average value of each state shows a very similar trend, although the increment
of SD and ST causes the decline of average SMP1 (Supplementary Fig. 6). According
to the suggestions of the reviewer, to clarify the influence of peak value of SMP1 on
the recognition accuracy, additional training simulations of four sampling conditions
have been conducted by the same dual-feature strategy, as shown in Fig. R10. The
convergency trends of recognition accuracy are similar, which indicates that the
sampling conditions have a negligible influence on the recognition simulation results.
The possible reason is that the short-term memory of the device is a gradual process,
and the sampled analog values increase or decrease synchronously. Besides, the readout
network contains only one matrix to multiple with the reservoir analog values and
utilizes the Softmax function to generate final outputs, diluting the differences in the
SMP1 absolute values. These comparison results demonstrate that the photo-synapse
reservoir could benefit from an elastic read time (sampling condition) of the analog
current.

**Fig. R10 Accuracy convergence curves of the training process with different SD**

**and ST sampling conditions. a** SD=10 ms, ST=10 ms; **b** SD=20 ms, ST=20 ms; **c**
ST=0 ms, ST=20 ms; **d** SD=0 ms, ST=30 ms. Even a large SMP1 range indicates a high
recognition capability, the trends of accuracy convergency under different sampling
conditions are similar. The possible reason is that the Softmax function dilutes the
differences in the SMP1 absolute values.

**Comment 4:** The dual-feature strategy sharps the feature of various inputs and
improves the recognition accuracy. But it also increases the burden of the readout layer.
Can the authors comment this effect on the overall performance?

**Reply to Comment 4:** We thank the reviewer for his/her approval that the dual-feature
strategy sharps the feature of various inputs and improves the recognition accuracy with
respect to the single-feature strategy. About the increment of the burden of the readout
layer, it is a typical dilemma between the system recognition accuracy and the hardware
consumption. Obviously, the dual-feature strategy system increases the hardware
burden of the RC system. This topic of the dilemma between system recognition rate
and hardware burden deserves further study.

From the perspective of high recognition accuracy, increment in hardware burden to a
certain extent is acceptable. The typical two-terminal structure of memristor is highly
CMOS compatible and ensures its unparalleled advantage in high density integration.
Memristor chips in the scales beyond Mb have already been broadly reported^{R17, 18}.
Therefore, even only a 32×32 memristor array is utilized in this work, large array will
support the dual-feature strategy to facilitate a high recognition accuracy. At the same
time, with the development of energy-efficient memristor array, the whole system
would perform a lower power consumption.

In addition, optimization of the reservoir in the single-feature strategy could be another
scheme to alleviate the dilemma. The low training speed in the single-feature strategy
is mainly caused by the overlaps between the feature value distributions. Therefore,
optimization of the reservoir architecture to sharpen the feature value distribution will
also improve the final training result, making it comparable to the dual-feature strategy.

**Comment 5:** In Supplementary Fig. 11, pulse stimulations for increment and decrement

of memristor conductance are missing. Also, the description of the training method of
 the memristor array is unclear in method section. The authors should make it more clear.

**Reply to Comment 5:** We thank the reviewer for this helpful comment.

We have modified the relevant figures and captions about the memristor array
 operations and characteristics in the revised manuscript:

(main text, Fig. 4a)

“

”

(SI, Revised Supplementary Fig. 12)

“

**Supplementary Fig. 12 Basic operations and resistance/conductance**
**characteristics of the memristor in the array. a** Operation parameters (left) and the
I-V characteristics (right) under DC double sweep mode of one typical memristor.
When the source line (SL) is grounded and the bit line (BL) is fixed at a certain voltage,
the DC voltage on the word line (WL) conducts double-sweep from 0 to 2.5 V to SET
and 0 to -2.5 V to RESET. The resistance state can be well modulated by different
compliance currents determined by the bias of BL. **b** Operation parameters of the pulse
SET (left) and pulse RESET (middle) and the gradual conductance modulation for 5
cycles under successive stimulations (right) of one typical memristor. In the
conductivity rising stage, only the pulse SET operations are implemented, in which the
bit line voltage increases from 1 to 2 V with a step of 0.01 V. While in the conductivity
decline stage, each conductance state is modulated by a couple of pulse RESET and
pulse SET: first, a RESET operation is conducted to erase the conductance; then, a pulse
SET is applied, in which the bit line voltage decreases from 2 to 1 V with a step of -
0.01 V. The conductance value could be repeatedly regulated within approximately
0-300 μS .”

For the training of the memristor array, we utilized offline training method to update
the weights (conductance) matrix of the array. Once the software simulation was
completed, the weights of the whole array (400 memristor devices) were updated
referring to the simulation results, column by column. The operation parameters are
illustrated in the revised Supplementary Fig. 12.

According to the reviewer’s comments, we have added the more detailed memristor
modulation and training methods into the revised main text (**Methods, Network**
**training**):

“Each differential pair in the memristor array represents a single weight of the neural
network. Transistors are used for device addressing and crosstalk current suppression.
As for the training of the memristor array, we utilized an offline training method to
update the weight (conductance) matrix of the array. Once the software simulation is
completed, the weights of the whole array (400 memristor devices) are updated by
referring to the simulation results, column by column. To SET a selected column, all

source lines (in blue, in Fig. 4a) were floated, except the selected one, which was
grounded. All word lines (in red) were biased at the same SET voltages.”

**Comment 6:** “differentcomplicance” in Supplementary Fig. 9 should be “different
compliance”. Please check the English throughout the manuscript.

**Reply to Comment 6:** We thank the reviewer for this helpful comment. This typo has
been corrected. According to the reviewer’s comments, the English expression of the
full text has been checked and polished. All the revisions about typos and grammar in
the revised main text and SI are highlighted in yellow.

**References**

- R1. Zhong, Y. et al. Dynamic memristor-based reservoir computing for high-efficiency temporal
signal processing. *Nat. Commun.* **12**, 408 (2021).
- R2. Milano, G. et al. In materia reservoir computing with a fully memristive architecture based on
self-organizing nanowire networks. *Nat. Mater.* **121**, 195–202 (2021).
- R3. Moon, J. et al. Temporal data classification and forecasting using a memristor-based reservoir
computing system. *Nat. Electron.* **2**, 480-487 (2019).
- R4. Appeltant, L. et al. Information processing using a single dynamical node as complex system. *Nat.*
*Commun.* **2**, 468 (2011).
- R5. Midya, R. et al. Reservoir computing using diffusive memristors. *Adv. Intell. Syst.* **1**, 1900084
(2019).
- R6. Sun, L. et al. In-sensor reservoir computing for language learning via two-dimensional memristors.
*Sci. Adv.* **7**, eabg1455 (2021).
- R7. Du, C. et al. Reservoir computing using dynamic memristors for temporal information processing.
*Nat. Commun.* **8**, 2204 (2017).
- R8. Wang, Y. et al. MXene-ZnO memristor for multimodal in-sensor computing. *Adv. Funct. Mater.*
**31**, 2100144 (2021).
- R9. Du, W. et al. An optoelectronic reservoir computing for temporal information processing. *IEEE*
*Electron Device Lett.* **43**, 406 (2022).
- R10. Liang, H. et al. Flexible X-ray detectors based on amorphous Ga₂O₃ thin films. *ACS Photonics* **6**,
351-359 (2018).
- R11. Antonello Tebano et al. Room-temperature giant persistent photoconductivity in SrTiO₃/LaAlO₃
heterostructures. *ACS Nano* **6**, 1278-1283 (2012).
- R12. Jang, J. T. et al. Study on the photoresponse of amorphous In-Ga-Zn-O and zinc oxynitride
semiconductor devices by the extraction of sub-gap-state distribution and device simulation. *ACS*

*Appl. Mater. Interfaces* **7**, 15570-15577 (2015).

R13. Cui, S. et al. Room-temperature fabricated amorphous Ga₂O₃ high-response-speed solar-blind
photodetector on rigid and flexible substrates. *Adv. Opt. Mater.* **5**, 1700454 (2017).

R14. Chen, K.-Y. et al. The effect of oxygen vacancy concentration on indium gallium oxide solar
blind photodetector. *IEEE Trans. Electron Devices* **65**, 1817-1822 (2018).

R15. Feng, Z. et al. Influence of annealing atmosphere on the performance of a β-Ga₂O₃ thin film and
photodetector. *Opt. Mater. Express* **8**, (2018).

R16. Qian, L. X. et al. Simultaneously improved sensitivity and response speed of β-Ga₂O₃ solar-blind
photodetector via localized tuning of oxygen deficiency. *Appl. Phys. Lett.* **114**, (2019).

R17. Cheng-Xin Xue et al. A 1Mb multibit reRAM computing-in-memory macro with 14.6ns parallel
MAC computing time for CNN Based AI edge processors. *2019 IEEE International Solid-State
Circuits Conference*, (San Francisco, 2019).

R18. Pulkit Jain, U. A. et al. A 3.6Mb 10.1Mb/mm² embedded non-volatile reRAM macro in 22nm
FinFET technology with adaptive Forming/Set/Reset schemes yielding down to 0.5V with
sensing time of 5ns at 0.7V. *2019 IEEE International Solid-State Circuits Conference*, (San
Francisco, 2019).

**The list of the corrections of NCOMMS-22-15956:**

After carefully considering the reviewers' valuable comments/suggestions, we revised
the manuscript in detail. All the revised text, data, and notes in the manuscript are
highlighted in yellow. The list of the corrections is recorded as follows **by the order**
**of occurrence** in the revised manuscript:

Former Version	Revised Version
(Title) In-sensor reservoir computing system for in-situ latent fingerprint recognition with deep ultraviolet photo-synapses and memristor array	(Title) In-sensor reservoir computing system for latent fingerprint recognition with deep ultraviolet photo-synapses and memristor array
(main text, Introduction, Paragraph 2) Fortunately, a promising strategy of in-sensor RC computing based on optoelectronic devices has been proposed for temporal sensory information processing and verified with the assistance of system simulation^{21, 22}. Nevertheless, the waveband utilized in these works is not suitable for DUV detection.	(main text, Introduction, Paragraph 2) Fortunately, a promising strategy of in-sensor RC based on optoelectronic devices has been proposed for temporal sensory information processing and verified with the assistance of system simulation^{21, 22}. In order to fulfill the in-sensor applications, the optoelectronic devices should be marked by the properties of nonlinearity response, short-term memory (STM), multiple states and stability. Nevertheless, the waveband utilized in above works is not suitable for DUV detection.
(main text, Fig. 2)  Fig. 2 PPC effect and synaptic behavior of the a-GaO_x DUV sensor.	(main text, Fig. 2)  Fig. 2 PPC effect and synaptic behavior of the a-GaO_x DUV sensor.
(main text, Nonlinear mapping of 4-bit inputs of the a-GaOx DUV reservoir, Paragraph 2)	(main text, Nonlinear mapping of 4-bit inputs of the a-GaOx DUV reservoir, Paragraph 2)

For illustrating the feature sampling, the I-t curves of three representative inputs of “0001” (in blue), “0011” (in red), and “1101” (in purple) of the a-GaOx reservoir are exhibited in Fig. 3b. Based on the conspicuous difference, each pixel sequence can be distinguished by current sampling to realize feature extraction.	To illustrate the feature sampling, the I-t curves of three representative inputs of “0001” (in blue), “0011” (in red), and “1101” (in purple) of the a-GaOx reservoir are exhibited in Fig. 3b. Although the last pulses are all “1”, their decay processes after the input sequences are different. Therefore, the final state of the reservoir not only relates to the last input, but also depends on its real-time state, indicating the lateral connections in such an a-GaOx reservoir^{21,22}. Based on the conspicuous difference, each pixel sequence can be featured by current sampling to realize feature extraction.
(main text, Nonlinear mapping of 4-bit inputs of the a-GaOx DUV reservoir, Paragraph 3) Consequently, the feature space based on nonlinear photoresponse configures the classification process of the reservoir, reducing the dimensionality of raw data from 4-bit digital inputs to 2 analog outputs that serve as the inputs of the linear readout layer^{49, 50}.	(main text, Nonlinear mapping of 4-bit inputs of the a-GaOx DUV reservoir, Paragraph 3) Consequently, the feature space based on nonlinear photoresponse configures the classification process of the reservoir, reducing the dimensionality of raw data from 4-bit digital inputs to 2 analog outputs that serve as the inputs of the linear readout layer^{49, 50}. The energy consumption per pulse operation of the optoelectronic reservoir can be estimated to be $E=20 \text{ nA} \times 1 \text{ V} \times 25 \text{ ms}=0.5 \text{ nJ}$, indicating that the reservoir architecture possesses potential energy-efficient characteristic.
(main text, Fingerprint recognition with fully-hardware DUV in-sensor RC system, Paragraph 1) Thus, a dual-feature strategy is employed, and only a dimensionality-reduced 40×5 weight matrix needs to be trained for each fingerprint image.	(main text, Fingerprint recognition with fully-hardware DUV in-sensor RC system, Paragraph 1) Thus, a dual-feature strategy is employed, and only a dimensionality-reduced 40×5 weight matrix needs to be trained for each fingerprint image. As an example, a recognition accuracy for unseen fingerprint images has been simulated to be around 92% based on this dual-feature strategy, where the expanded sample amounts of the fingerprint images ensure the high recognition accuracy (see Supplementary Fig. 11).

(main text, Fingerprint recognition with fully-hardware DUV in-sensor RC system, Paragraph 3) Three situations, full-precision (double-precision floating-point) simulation, limited-precision (32-bit fixed-point quantization) simulation, and hardware experiment, are considered for comparison. As shown in Fig. 4f, even recognition accuracies in all situations deteriorate with the increment of noise level, the recognition accuracy of hardware experiment still maintains above 90% under 15% noise level. Therefore, the fully-hardware DUV in-sensor RC system based on a-GaOx photo-synapse has promising potential to be competent for high-precision in-situ DUV fingerprint recognition tasks. It should be noted that the increase of resistive states of the memristor device could significantly improve the system robustness⁵¹.	(main text, Fingerprint recognition with fully-hardware DUV in-sensor RC system, Paragraph 3) Three situations, full-precision (double-precision floating-point) simulation, limited-precision (32-bit fixed-point quantization) simulation, and hardware experiment, are considered for comparison. As shown in Fig. 4f, recognition accuracies in all situations remain comparable under $\leq 3\%$ noise level and deteriorate asynchronously with its increment. The limited resistive states of the memristor device and the amplification of non-ideal factors (e.g., device-to-device and cycle-to-cycle variations, discreteness of operations, etc.) under high-level noise dominate the relatively quick deterioration in the hardware situation. Therefore, the improvement of resistive states and uniformity of the memristor devices could further improve the system robustness⁵¹. It is noteworthy that the recognition accuracy of the hardware experiment still maintains above 90% under 15% noise level.
(main text, Methods, Basic memristor array operations) Transistors are used for device addressing and crosstalk current suppression. For weight programming, the memristor array was programmed column by column. To SET a selected column, all source lines (in blue, in Fig. 4a) were floated, except the selected one, which was grounded.	(main text, Methods, Basic memristor array operations) Transistors are used for device addressing and crosstalk current suppression. As for the training of the memristor array, we utilized an offline training method to update the weight (conductance) matrix of the array. Once the software simulation is completed, the weights of the whole array (400 memristor devices) are updated by referring to the simulation results, column by column. To SET a selected column, all source lines (in blue, in Fig. 4a) were floated, except the selected one, which was grounded.
(main text, Methods, Network training)	(main text, Methods, Network training)

The fully-connected network was trained by MATLAB Deep-learning Toolbox, with softmax activation function and back-propagation algorithm.

The fully-connected network was trained by the MATLAB Deep-learning Toolbox, utilizing the Softmax output function and the logistic regression to supervise the learning.

(SI, Supplementary Fig. 2)

Supplementary Fig. 2 a-GaO_x device and its nonlinear photoresponse. **a** Schematic diagram of the cross-section structure of the a-GaO_x device. **b** Nonlinear dependence of ΔI on DUV light pulse width (25 ms) under various power densities.

(SI, Supplementary Fig. 2)

Supplementary Fig. 2 a-GaO_x device and its nonlinear photoresponse. **a** Schematic diagram of the cross-section structure of the a-GaO_x device. **b** Nonlinear dependence of ΔI on DUV light pulse width (25 ms) under various power densities.

(SI, Supplementary Fig. 7)

Supplementary Fig. 7 Statistic data of SMP1 of the 16 inputs from stochastically selected 20 a-GaO_x photo-synapse devices. The

(SI, Supplementary Fig. 7)

Supplementary Fig. 7 Statistic data of SMP1 of the 16 inputs from stochastically selected 20 a-GaO_x photo-synapse devices. The sampling parameters are fixed at SD=0 ms and ST=10 ms. A similar distribution trend indicates that all devices

sampling parameters are fixed at $SD=0$ ms and $ST=10$ ms. A similar distribution trend indicates that all devices exhibit decent classification performances.

exhibit decent classification performances.

None

(SI, Supplementary Fig. 8)

Supplementary Fig. 8 Schematic diagram of the parallel time-delayed reservoir network as a demonstration of our work. The image is divided suitably then input into the reservoirs in parallel. The virtual nodes of each reservoir are coupled with a time interval θ . For the designed readout network, only the last 1 or 2 nodes of each reservoir are utilized to construct the output vector.

None

(SI, Supplementary Fig. 11)

Supplementary Fig. 11 Recognition simulation of the unseen fingerprint images. a Expansion of the data set of the fingerprints from 40 to 400 images by introducing one random noise pixel in each

original image for 10 times (taking the C-1 image in Supplementary Fig. 10 as an example), owing to the finite scale of the FVC 2002 database. 80% of the fingerprint images were set as the training set and the other 20% as the test set (namely the unseen images). **b** Accuracy convergency of the training process within 1000 epochs. Considerable recognition accuracy can be achieved upon certain training epochs. **c** Confusion matrix of the fingerprint recognition with the unseen images as the test set. The test accuracy is extracted to be 92.5%.

(SI, Supplementary Fig. 10 and 11)

Supplementary Fig. 10 I-V characteristics of SET and RESET characteristics of one typical memristor in the array. The resistance state can be well modulated by different compliance current.

Supplementary Fig. 11 Gradual

(SI, Supplementary Fig. 12)

Supplementary Fig. 12 Basic operations and resistance/conductance characteristics of the memristor in the array. **a** Operation parameters (left) and the I-V characteristics (right) under DC double sweep mode of one typical memristor. When the source line (SL) is grounded and the bit line (BL) is fixed at a certain voltage, the DC voltage on the word line (WL) conducts double-sweep from 0 to 2.5 V to SET and 0 to -2.5 V to RESET. The resistance state can be well modulated by different compliance currents determined by the bias of BL. **b** Operation parameters of the pulse SET (left) and pulse RESET (middle) and the gradual conductance modulation for 5 cycles under successive stimulations (right) of one typical memristor. In the conductivity rising stage, only the pulse SET

conductance modulation for 5 cycles under successive pulse stimulations based on one typical memristor in the array.	operations are implemented, in which the bit line voltage increases from 1 to 2 V with a step of 0.01 V. While in the conductivity decline stage, each conductance state is modulated by a couple of pulse RESET and pulse SET: first, a RESET operation is conducted to erase the conductance; then, a pulse SET is applied, in which the bit line voltage decreases from 2 to 1 V with a step of -0.01 V. The conductance value could be repeatedly regulated within approximately 0-300 μS.
--	--

627

628

REVIEWER COMMENTS

Reviewer #1 (Remarks to the Author):

The authors have rigorously addressed all the feedback provided by the reviewers.

I would urge that authors address this one part clearly:

If there is excellent yield for the memristive crossbar, why AMP task results are achieved via simulation? or in other words,

based on the response, the memristor-based RC network requires optimization of read-out circuit, ADC, control unit, etc.

It is unclear if these parts of the system are on silicon or not. Further, it is important to note how would the system performance change if one considered the ADC error and overhead of peripherals.

Reviewer #2 (Remarks to the Author):

The authors have made satisfactory revisions according to the reviewers' suggestions. I would recommend it to be published in its present form. Regarding the latest progress on in-sensor computing, the authors may refer to *Advanced Materials*, 2022, 2203830.

Response to Reviewer's Comments

This Response Letter is regarding a former manuscript submitted to *Nature Communications*, entitled “*In-sensor reservoir computing system for latent fingerprint recognition with deep ultraviolet photo-synapses and memristor array*” by Zhongfang Zhang et al. (NCOMMS-22-15956A). We would like to express our special thanks for the affirmation from the reviewers about the revised version of this work. In the following response, the original comments are in black font, our responses are in blue font, and changes in the revised main text are highlighted in yellow.

I. Comments from Reviewer 1

Overall Comment:

The authors have rigorously addressed all the feedback provided by the reviewers.

I would urge that authors address this one part clearly:

If there is excellent yield for the memristive crossbar, why AMP task results are achieved via simulation? Or in other words, based on the response, the memristor-based RC network requires optimization of read-out circuit, ADC, control unit, etc.

It is unclear if these parts of the system are on silicon or not. Further, it is important to note how would the system performance change if one considered the ADC error and overhead of peripherals.

Reply to Comment: We thank the referee for the precious time and constructive comments on our manuscript.

We guess the reviewer means that the AMP task of memristor array readout is simulated. In our work, the peripheral circuits were integrated on a printed circuit board, including ADC, TIA, and control unit, as shown in Fig. R1.

Fig. R1 The integrated control units and peripheral circuits on the test board.

The memristor array size is 32×32 , which is not adapted to directly construct a network with 40 inputs. Thus, a block processing method is utilized on the readout array: the 40×10 network is divided into 30×10 and 10×10 parts and assigned separately in the array, and then the currents of each two correlated columns are manually added for further processing by a computer (utilizing Softmax function via simulation). It should be noted that increasing the size of the array will make the readout of the network with only one operation. Even so, the simulation only exists in training process, and the inference is based on hardware data. In Fig. 4d and Fig. 4e, the actual conductance values of the memristor hardware were multiplied by a constant (1.25×10^4), to make the hardware and simulation results share the same color bar for better comparison. This does not mean that the AMP in the readout process relies on simulation. To avoid conflict, we added descriptions about the multiplication constant in the caption of Figure 4d:

“

Fig. 4 Fingerprint recognition based on hardware DUV in-sensor RC system. **d** The colormaps and **e** statistic histograms of the 40×5 weights of the simulation and

hardware experiment, respectively. The actual conductance values read from hardware were multiplied by a constant of 1.25×10^4 for better comparison with the simulated weights.”

The read-out circuit, ADC, control unit, *etc.*, are based on silicon. We thank the reviewer for the constructive suggestion to achieve a compact integration of the whole in-sensor computing system. Both the photo-synapse devices and peripherals deserve further optimization in our future work.

There are already mature techniques of ADC and peripherals parts to meet the requirements of commercialized applications. By conducting quantization of the input and output values of the memristor array, we have conducted simulations of the influences of ADC precisions on the inference results, as shown in Fig. R2. As long as there are no significant errors during multiple operation cycles, the performance will be well preserved even when the ADC precision is down to 8 bits. Compared to the errors of photo-synapse and memristor (*e.g.*, device-to-device and cycle-to-cycle variations, *etc.*), the circuit parts have relatively little impact on the performance in this in-sensor RC system.

Fig. R2 The influence of ADC precision on test accuracy with increasing image noise level. Quantization from 8 bits to 64 bits of the input and output values of the memristor array simulated the ADC precision. The influence of ADC precision on the recognition results is far inferior to that of image noise.

In addition, the peripheral overhead mainly comes from ADC, which means that the lower precision of ADC can reduce the hardware overhead, indicating a typical trade-

off in system construction. It has been proven that the ADC precision reduction does not seriously affect the system performance in this work, therefore it is better to use low-precision ADC to construct the system. But if the accuracy requirement is very tough, using low-precision ADC may deteriorate the performance, which means that the accuracy and overhead need to be considered comprehensively.

II. Comments from Reviewer 2

Overall Comment:

The authors have made satisfactory revisions according to the reviewers' suggestions. I would recommend it to be published in its present form. Regarding the latest progress on in-sensor computing, the authors may refer to *Advanced Materials*, 2022, 2203830.

Reply to Comment: We thank the referee for the precious time and positive comments on our manuscript. The suggested paper has been added into the corresponding location and the number of the references is updated accordingly in the revised manuscript:

(main text, **Introduction, Paragraph 1**)

“In addition, these systems utilize additional optical filters for charge-coupled devices (CCDs) and complementary metal-oxide-semiconductor (CMOS) image sensors, increasing the complexity of the entire system for latent fingerprint identification¹⁵⁻¹⁷.”

(References)

“17. Wan, T. et al. In-sensor computing: materials, devices, and integration technologies. *Adv. Mater.* **2022**, e2203830 (2022).”

REVIEWERS' COMMENTS

Reviewer #1 (Remarks to the Author):

The authors have updated the manuscript with the suggestions provided. The manuscript can be accepted for publication.

Manuscript ID: NCOMMS-22-15956B

Response to Reviewer's Comments

This Response Letter is regarding a former manuscript submitted to *Nature Communications*, entitled “*In-sensor reservoir computing system for latent fingerprint recognition with deep ultraviolet photo-synapses and memristor array*” by Zhongfang Zhang *et al.* (NCOMMS-22-15956B). We would like to express our special thanks for the affirmation from the editor and reviewers about the revised version of this work.

Comments from Reviewer 1

Overall Comment:

The authors have updated the manuscript with the suggestions provided. The manuscript can be accepted for publication.

Our response: We sincerely thank the reviewer for the positive comments that our work is acceptable.

We also sincerely thank the positive assessments from reviewer 2 in the last revision process that this work is commendable and could provide more insights.

Thanks for their recommendation of this manuscript for acceptance in *Nature Communications*.